# Protocol of a mixed (Retrospective and Prospective) longitudinal observational study of amiodarone for chronic Chagas cardiomyopathy (AMIO-CHAGAS study): Investigating immunomodulatory and trypanocidal effects

Juliana Magalhães Chaves Barbosa[1,2], Amanda Faier-Pereira[3],
Alejandro Marcel Hasslocher-Moreno[4], Andréa Rodrigues Costa[4],
Evelyn Nunes Goulart da Silva Pereira[2], Fernanda de Souza Nogueira Sardinha Mendes[4],
Gilberto Marcelo Sperandio da Silva[4], Luiz Henrique Conde Sangenis[4],
Marcelo Teixeira Holanda[4], Mauro Felippe Felix Mediano[4], Roberto Magalhães Saraiva[4],
Sérgio Salles Xavier[4], Otacílio da Cruz Moreira[3], Anissa Daliry[2], Henrique Horta Veloso[4],
Kelly Salomão[1]*

**1** Cell Biology Laboratory, Oswaldo Cruz Institute, Oswaldo Cruz Foundation, Rio de Janeiro, Rio de Janeiro, Brazil, **2** Laboratory of Clinical and Experimental Physiopathology, Oswaldo Cruz Foundation, Rio de Janeiro, Rio de Janeiro, Brazil, **3** Platform of Molecular Analysis, Laboratory of Molecular Virology and Parasitology, Oswaldo Cruz Foundation, Rio de Janeiro, Rio de Janeiro, Brazil, **4** Evandro Chagas National Institute of Infectious Diseases, Oswaldo Cruz Foundation, Rio de Janeiro, Rio de Janeiro, Brazil

☯ These authors contributed equally to this work.
* ks@ioc.fiocruz.br

## Abstract

### Background

Chronic Chagas cardiomyopathy (CCC) is the most significant clinical manifestation of Chagas disease (CD) due to its impact on morbidity and mortality. Cardiac manifestations include dilated cardiomyopathy, with ventricular dysfunction, heart failure, cardiac arrhythmias and cardioembolic events. Amiodarone (AMIO) is the most commonly used antiarrhythmic drug in CCC primarily prescribed for patients with severe conduction system abnormalities. Despite preliminary evidence, no clinical study has specifically evaluated the immunomodulatory and trypanocidal effects of AMIO in patients with CCC. The main objective is to evaluate the association of AMIO therapy with serological reactivity to anti-*T. cruzi* and the inflammatory profile of patients with CCC.

### Methods/Design

This is a mixed (retrospective and prospective) longitudinal observational study enrolling 90 patients who will be recruited from a reference center for CD in Brazil. Patients with CCC receiving AMIO will be compared with those not receiving the drug. AMIO was prescribed at the discretion of the treating physician, independently of the present study. Eligible

**Data availability statement:** No datasets were generated or analysed during the current study. All relevant data from this study will be made available upon study completion.

**Funding:** The author(s) received no specific funding for this work.

**Competing interests:** The authors have declared that no competing interests exist.

participants will be invited to participate during routine medical appointments. Clinical data, including electrocardiographic and echocardiographic parameters obtained during routine follow-up, will be retrospectively extracted from medical records. Blood samples will be collected at predetermined time points. The primary endpoint will be the inflammatory profile and serological reactivity to anti-*T. cruzi* in CCC patients. Secondary endpoints will include: (**1**) cardiovascular events; (**2**) CCC progression; (**3**) implantable cardioverter-defibrillator (ICD) implantation; (**4**) heart transplantation; (**5**) all-cause death and (**6**) a composite endpoint. The association between AMIO use and longitudinal changes in continuous variables will be determined using mixed linear models. Cox regression analysis will be used to investigate the relationship between AMIO use and selected outcomes (cardiovascular events, CCC progression, and death), adjusted for potential confounders. Hazard ratios (HRs) with 95% confidence intervals (CIs) will be reported.

## Discussion

Given the limited literature on the potential trypanocidal and immunomodulatory effects of AMIO in CCC, the findings of this study may provide new insights into the therapeutic potential of AMIO beyond its established antiarrhythmic properties.

## Background

Chagas disease (CD) is an anthropozoonosis caused by the parasite *Trypanosoma cruzi* (*T. cruzi*), affecting approximately 8 million people worldwide and resulting in over 14,000 deaths annually. About 70 million people globally live in areas at risk of *T. cruzi* infection [1–4]. Cardiac chronic CD is responsible for the most significant morbidity and mortality among parasitic diseases and imposes the highest economic and health burden from parasitic diseases in the Western Hemisphere [5,6], leading to a substantial financial burden on health-care systems, with estimated health-care costs of US$690 million [7–9].

In recent decades, CD has become a global health problem due to increased migration from endemic to non-endemic areas such as Canada, the USA, Europe, Australia, and Japan. In these regions, *T. cruzi* transmission occurs mainly through vertical transmission from mother to child [1,9–11]. It is estimated that 2.8% of Latin American immigrants living in Europe are infected with *T. cruzi*, and 90% of these individuals remain undiagnosed [11].

In chronic Chagas cardiomyopathy (CCC), the replacement of cardiac fibers with connective tissue leads to loss of contractile capacity and ventricular remodeling, resulting in dilation of the cardiac chambers. These morphophysiological changes cause a gradual loss of ejection capacity, contributing to heart failure (HF) [6]. In addition to HF, arrhythmia is another important manifestation in CD patients with cardiac involvement, causing palpitations, dizziness, dyspnea, lipothymia, syncope and sudden death [12]. Arrhythmias and conduction disorders can occur in both the acute and chronic phases [13]. However, these manifestations are more frequent during the chronic phase and at any stage of the disease [14].

Amiodarone (AMIO) is the most commonly used antiarrhythmic drug in CCC primarily prescribed for abnormalities in the cardiac conduction system [12], and is considered effective in reducing both atrial and ventricular arrhythmias [15,16]. This drug is a class III antiarrhythmic according to the Vaughan Williams classification, and its mechanisms of action include inhibition of potassium, calcium, and sodium channels, thereby prolonging the plateau of the cardiac cell action potential [17,18]. Due to the scarcity of studies involving individuals with CD, the use of AMIO in patients with CCC is based on an expected benefit that should outweigh its adverse effects, such as pulmonary toxicity, polyneuropathy, gastrointestinal discomfort, bradycardia, hepatotoxicity, thyroid dysfunction, and ocular complications [15,19,20]. Therefore, this study will evaluate the potential benefits associated with the use of AMIO in patients with CCC, including possible trypanocidal and immunomodulatory effects of this antiarrhythmic drug.

## Methods/Design

### Study setting

Individuals followed at the Evandro Chagas National Institute of Infectious Diseases (INI) of the Oswaldo Cruz Foundation (Fiocruz) will be invited during a medical appointment to participate in the AMIO-CHAGAS study. INI-Fiocruz, located in Rio de Janeiro, Brazil, is a national referral center for the treatment and research of infectious and tropical diseases within the framework of the Brazilian Unified Health System (SUS). INI-Fiocruz receives patients from across the country and provides comprehensive, multidisciplinary care for individuals with CD [21]. Patients currently living in the state of Rio de Janeiro with suspected or confirmed CD are referred to INI for diagnostic investigation or specialized healthcare. Many of these patients are migrants from endemic rural areas. The INI staff includes infectious disease specialists, cardiologists, gastroenterologists, nurses, nutritionists, pharmacists, and physical therapists or exercise physiologists. Patients routinely undergo blood tests, echocardiography (ECHO), electrocardiogram (ECG), 24-hour Holter monitoring, exercise testing, and cardiopulmonary exercise testing, all of which are available at this center. Data from medical records will be collected using standardized protocols. In the event of inconsistencies, records will be reviewed independently by two examiners. Only data with confirmed consistency will be included in the analyses.

### Study design

This is a mixed (retrospective and prospective) longitudinal observational study designed to evaluate the association between chronic use of amiodarone (AMIO) and the following:

(I)   **Primary outcome:** inflammatory profile, serologic reactivity to anti-*T. cruzi* antibody, and parasitic load;

(II)  **Secondary outcomes:** cardiovascular events (new sustained atrial arrhythmias, new sustained or non-sustained ventricular tachyarrhythmias, stroke, heart failure, atrioventricular block, and cardiovascular hospital admissions), CCC progression, internal cardioverter-defibrillator (ICD) implantation, heart transplantation, all-cause death, and a composite endpoint (including the aforementioned cardiovascular events, disease progression, ICD implantation, heart transplantation, and death).

Patients with CCC who take AMIO (exposed group) will be compared with those who do not take this drug (non-exposed group). To minimize potential confounding, patients will be matched based on the stage of CCC according to the Brazilian Consensus on Chagas Disease (A and B1 *versus* B2 and C), age (<50 years *versus* >50 years), and sex (female *versus* male) [15].

Clinical and laboratory data including blood tests, ECG, transthoracic two-dimensional echocardiography (ECHO), and 24-hour Holter monitoring, will be evaluated at three different time points:

(i) **Retrospective data**: from up to 3 years before the start of the study;

(ii) **Baseline/recruitment**: at the time of inclusion in the study;

(iii) **Prospective data**: during the 2 years of follow-up after recruitment;

The duration of the study, totaling five years of clinical follow-up, was established considering the CCC progresses at a rate of approximately 2% per year [22]. At recruitment, blood samples will be collected for biochemical and molecular analyses to obtain information related to the study's primary outcome (Table 1) (Fig 1).

## Study population

Participant recruitment began on March 7, 2023, and is expected to be completed by December 2025. Participants are being recruited sequentially during their routine medical visits. Patients using AMIO are identified from the database of individuals monitored by the Clinical Research Laboratory on Chagas Disease at INI-Fiocruz. Only patients who have been receiving AMIO for at least three months will be included, ensuring adequate tissue impregnation with a cumulative dose of 10 g [23].

Inclusion criteria are as follows:

1. Confirmed diagnosis of CD, based on two serological tests with different methodologies (enzyme-linked immunosorbent assay [ELISA] and chemiluminescence) [14,15];

2. Individuals over 18 years of age;

3. Patients classified within the following CCC stages: A, B1, B2, and C [15].

Exclusion criteria are as follows:

1. Co-infection (HIV, hepatitis B and C, HTLV, Hansen's disease, leishmaniasis and paracoccidioidomycosis);

2. Immunosuppression of any nature;

**Table 1. AMIO-CHAGAS's study Chronogram: Enrollment, Interventions and Outcome Assessments.**

| | | | Retrospective data | Recruitment | Prospective data |
|---|---|---|---|---|---|
| | | | 2020 −2022 | 2023 - 2025 | 2025 - 2027 |
| **Primary outcome:** | **Immunomodulatory effect** | Gene expression of CCC progression biomarkers in PBMCs | – | x | – |
| | | Determination of serum levels of cytokines and nitric oxide | – | x | – |
| | **Trypanocidal effect** | Determination of anti-*T. cruzi* antibody titers | – | x | – |
| | | *T. cruzi*-antigen quantification | – | x | – |
| **Secondary outcomes:** | Cardiovascular events | New sustained atrial arrhythmias and new sustained or non-sustained ventricular tachyarrhythmias, stroke, heart failure, atrio-ventricular block and cardiovascular hospital admissions | x | – | x |
| | Disease progression | Monitored by echocardiography (ECHO), electrocardiogram (ECG), 24-hour Holter | x | x | x |
| | ICD implantation | According to the medical prescription | – | – | x |
| | Heart transplantation | According to the medical prescription | – | – | x |
| | Death of all-cause | – | – | – | x |
| | Composite endpoint | – | – | – | x |

CCC: Chronic Chagas cardiomyopathy; ECG: Electrocardiogram; ECHO: Echocardiography; ICD: Implantable cardioverter-defibrillator; PBMCs: Peripheral blood mononuclear cells.

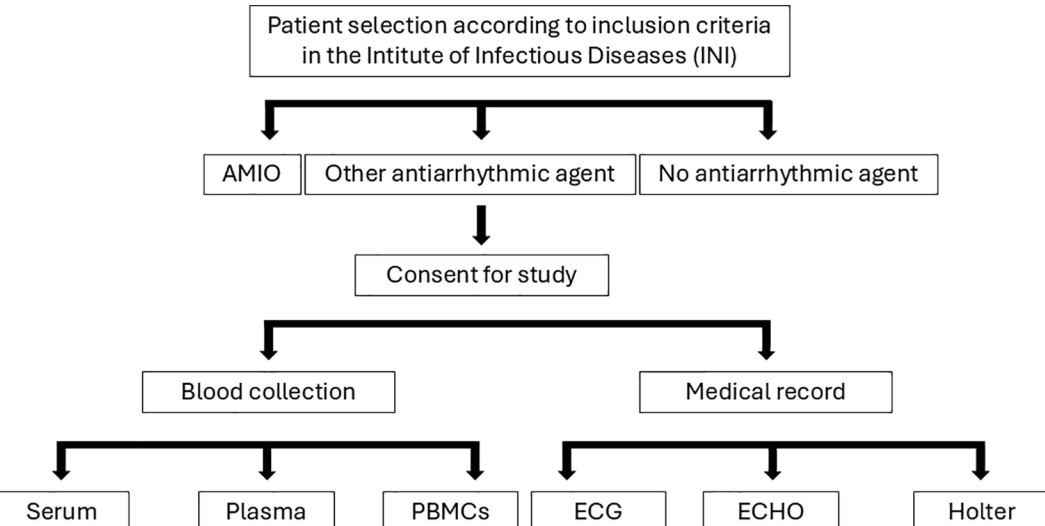

**Fig 1. Study flow diagram for data collection at recruitment.** ECG: Electrocardiogram; ECHO: Echocardiography; ICD: Implantable cardioverter-defibrillator; PBMCs: Peripheral blood mononuclear cells.

3. Cancer;

4. Patients with stage D of CCC will be excluded due to their complex clinical management, including frequent hospitalizations and medication adjustments, which could affect the study's feasibility.

Patients meeting the inclusion criteria will be invited to participate in the study, and written informed consent will be obtained from all individuals who agree to enroll.

Importantly, AMIO use was prescribed at medical discretion to treat atrial or ventricular arrhythmias [12], independently of this study, characterizing an observational study design.

## Blood collection and isolation of plasma, serum and peripheral blood mononuclear cells (PBMCs)

Venous blood will be collected by hospital staff during the routine consultation of patients enrolled in the study. Two samples of 5 mL each will be collected: one in BD Vacutainer® Plus Plastic Serum tubes and kept for at least 30 minutes at room temperature to obtain serum, and one in BD Vacutainer® Plus Plastic EDTA K3 tubes (Becton, Dickinson, New Jersey, USA), for the separation of PBMCs and plasma by Histopaque 1077 gradient (Sigma Aldrich™, St. Louis, USA) following the manufacturer's protocol (GE Healthcare). At the end of the separation procedure, the cells will be resuspended in 100 μL of TRI reagent (Sigma-Aldrich) and stored at −80º C until further analysis, such as plasma and serum. Biological samples will be stored during the study in accordance with established technical, ethical, and operational regulations, under institutional responsibility and the supervision of the principal investigator. Samples will be stored for up to five years after study completion and then discarded according to institutional biosafety regulations. During this period, samples may be used in additional subprojects subject to appropriate ethical approval.

## Analysis of gene expression of CCC progression biomarkers in PBMCs

The assessment of gene expression for key inflammatory and regulatory markers, including TNF, IFN-ɣ, IL-10, TGF-β, iNOS, and NF-κβ, in PBMCs provides valuable insights for monitoring systemic immune responses associated with chronic Chagasic cardiomyopathy (CCC) progression [24,25]. For this purpose, RNA will be extracted from collected

PBMCs using TRI reagent (Sigma-Aldrich). cDNA synthesis will be performed with the SuperScript III First-Strand Synthesis kit (Thermo Fisher Scientific, Waltham, MA, USA), according to the manufacturer's specifications. Gene expression will be measured by real-time quantitative PCR (RT-qPCR), using the following TaqMan gene expression assays: TNF (tumor necrosis factor; Assay ID: Hs00174128_m1), IFN-γ (interferon-gamma; Assay ID: Hs03044218_g1), IL-10 (interleukin; Assay ID: Hs00961622_m1), TGF-β (transforming growth factor beta; Assay ID: Hs00998133_m1), iNOS (inducible nitric oxide synthase; Assay ID: Hs01075529_m1) and NF-κβ (nuclear factor kappa β; Assay ID: Hs00765730_m1). Experiments will be conducted on a QuantiStudio 7 Pro Real-Time PCR System (Applied Biosystems), according to the manufacturer's instructions. Gene expression will be calculated by relative quantification, using the Expression Suite software (V1.3), on the Molecular Analysis Platform RTP09J (FIOCRUZ/RJ).

## Determination of serum levels of cytokines and nitric oxide

In addition to gene expression analysis of inflammatory markers in PBMCs, quantification of cytokine levels and nitric oxide in the serum of patients with chronic Chagasic cardiomyopathy (CCC) has previously been associated with the progression of symptomatic forms of the disease [26]. Accordingly, detection of IL-2, IL-17, IL-4, IL-6, IL-10, TNF, and IFN-γ will be performed using the Cytometric Bead Array Human Th1/Th2/Th17 kit (Becton Dickinson), following the manufacturer's protocol. Furthermore, nitrate and nitrite concentrations will be quantified in patient serum samples using the Griess reaction (Sigma-Aldrich), according to the manufacturer's specifications.

## Determination of anti-*T. cruzi* antibody titers and antigen quantification

Previous studies have indicated that reduced serological reactivity and lower anti-*T. cruzi* antibody titers may be associated with improved prognosis in CCC, as these changes have been correlated with improvements in patients' electrocardiographic profiles [27,28]. Based on this evidence, in the present study, the serological reactivity of anti-*T. cruzi* antibodies will be determined using the commercial Chagastest-Wiener kit (ELISA recombinant v.4.0), following the manufacturer's recommendations. Positive and negative controls will be included to ensure test functionality. The cut-off value will be calculated as the mean absorbance of the negative controls, with the indeterminate zone defined as within ±10% of the cut-off value. In parallel, antigen quantification will be performed to enable subsequent correlation between these two parameters. For the detection and quantification of *T. cruzi* in patient samples, the NAT Chagas kit (Nucleic Acid Test for Chagas Disease) (IBMP, Curitiba, Brazil) will be used. This kit, based on a duplex TaqMan assay targeting the nuclear satellite DNA of *T. cruzi* and an exogenous internal amplification control, offers a quantification range from $10^4$ to 0.5 parasite equivalents per milliliter of blood. The assays will be performed according to the manufacturer's instructions.

## Electrocardiogram, two-dimensional transthoracic echocardiogram, 24-hour Holter and blood pressure

All analyses will be performed annually by Lapclin-Chagas (INI), by qualified professionals. The 12-lead electrocardiogram will be conducted with patients at rest, using a long recording on D2 to assess arrhythmias. The presence or absence of electrocardiographic changes will be determined according to the criteria recommended by the II Brazilian Consensus on Chagas Disease [15]. Electrocardiographic analysis will be performed using the Holter system (portable three-channel recorder and analyzer; Cardio Light® and CardioSmart® 5.0, Cardio System, São Paulo, Brazil), with three channels connected continuously for 24 hours at a rate of 200 samples per second, to detect arrhythmias predictive of mortality and to obtain the arrhythmogenic profile [29,30]. Two-dimensional transthoracic echocardiography will be performed using a phased-array ultrasound system (Philips CVx, USA) equipped with an M4S transducer, following the recommendations of the World Heart Federation [31]. Left ventricular systolic ejection fraction (LVEF), as well as left ventricular diastolic and systolic volumes, will be quantified using Simpson's biplane method of discs. Assessments will also include left ventricular diastolic function, right ventricular (RV) systolic function, and strain parameters of the left atrium (LA), RV, and left ventricle (LV), to provide a comprehensive evaluation of cardiac mechanics. All measurements will be performed by experienced

cardiologists blinded to the clinical data to minimize bias. Blood pressure will be measured on the left arm with participants seated in a quiet room after 5 minutes of rest, using an Omron digital sphygmomanometer.

## Sample size

A minimum sample size of 90 volunteers was estimated, with 45 using AMIO and 45 not using AMIO. The sample size calculation for the primary endpoint was based on a study by Rodriguez-Angulo et al., 2017 [26] which evaluated differences in inflammatory markers in individuals with CD treated with AMIO (IFN-γ: 43.75±24.83; TNF-α: 27.08±13.81) compared to untreated individuals (IFN-γ: 84.37±35.08; TNF-α: 52.08±23.38). For the secondary endpoint, sample size calculation was based on a study by Saraiva et al., 2022 [22], which compared LVEF at different stages of CCC (LVEV; Stage A: 68.0±6.4; Stage B: 54.7±9.1; Stage C: 36.7±10.7). Sample size calculations assumed a two-sided significance level of α=0.05, a statistical power of 80% (1−β=0.8), and standard deviations and mean differences derived from these studies. The largest sample requirement among the two endpoints was adopted, and the sample size was increased by 20% to account for potential losses or refusals, resulting in a total of 90 participants.

## Statistical analysis

Descriptive statistics were used to summarize the characteristics of the study population. Continuous variables will be presented as medians and interquartile ranges (IQR), and categorical variables as absolute and relative frequencies. Missing data will be assessed for their pattern and mechanism (i.e., missing completely at random (MCAR), missing at random (MAR), or missing not at random (MNAR)). The primary approach to handling missing data will be determined once the dataset is available. Depending on the extent and nature of missingness, appropriate strategies may include multiple imputation, complete case analysis, or sensitivity analyses to evaluate the robustness of the findings. The Shapiro-Wilk test will be used to assess the normality of continuous variables. When distributions deviate from normality, appropriate transformations or non-parametric tests will be applied.

Comparisons of patient characteristics according to AMIO use will be evaluated using the independent Student's t-test for normally distributed continuous variables or the Mann-Whitney U-test for non-normally distributed data. Categorical variables will be compared using the chi-square test or Fisher's exact test, as appropriate. Correlations between continuous variables related to the primary outcome will be assessed using Pearson or Spearman correlation coefficients, depending on data distribution. AMIO use and longitudinal changes in continuous variables will be analyzed using mixed linear models, adjusted for the duration of amiodarone use. Analyses of time course outcomes (e.g., cardiovascular events, disease progression, all-cause mortality) will be conducted using Cox proportional hazards regression models adjusted for potential confounders such as age, sex, comorbidities and previous etiologic treatment. Hazard ratios (HRs) and their 95% confidence intervals (CI) will be reported. A two-sided p-value ≤ 0.05 will be considered statistically significant.

## Data collection

Data collection included clinical, immunological, molecular, and imaging parameters, categorized as either categorical or continuous variables. The data generated and analyzed during this study are available from the corresponding author upon reasonable request, subject to ethical and privacy considerations.
  **Continuous variables included:**

(i)  Gene expression levels of TNF, IFN-γ, IL-10, TGF-β, iNOS, and NF-κB in PBMCs, quantified by RT-qPCR and expressed as fold change (arbitrary units);

(ii) Serum cytokine concentrations—including IL-2, IL-4, IL-6, IL-10, IL-17, TNF, and IFN-γ—quantified in pg/mL using a Cytometric Bead Array (CBA);

(iii)  Serum levels of nitrate and nitrite (µM), determined by the Griess reaction;

(iv)  Anti-*Trypanosoma cruzi* IgG antibody titers, quantified by ELISA and reported as absorbance values;

(v)  Parasitic DNA load (parasite equivalents/mL), quantified by duplex TaqMan qPCR using the NAT Chagas kit;

(vi)  Left ventricular systolic ejection fraction (LVEF, %), as well as left ventricular diastolic and systolic volumes (mL), quantified using Simpson's biplane method of discs via two-dimensional transthoracic echocardiography;

(vii)  Left ventricular diastolic function (graded by established echocardiographic criteria), right ventricular (RV) systolic function (expressed as fractional area change or TAPSE in mm), and strain parameters (%) of the left atrium (LA), right ventricle (RV), and left ventricle (LV), evaluated to provide a comprehensive assessment of cardiac mechanics;

(viii) Electrocardiographic parameters: heart rate (beats per minute, BPM) and electrocardiographic intervals, including PR interval (ms), QRS duration (ms), QT interval (ms), and corrected QT interval (QTc, ms);

(ix)  Systolic and diastolic blood pressure (mmHg), measured on the left arm with participants seated in a quiet room after 5 minutes of rest, using an Omron digital sphygmomanometer;

**Categorical variables included:**

(i)  Echocardiographic parameters: preserved left ventricular systolic function, left ventricular akinesia, left ventricular hypokinesia, mitral regurgitation, tricuspid regurgitation, pulmonary insufficiency, aortic insufficiency, septal hypertrophy, thrombus observation and aneurysm observation.

(ii)  Electrocardiographic parameters: sinus bradycardia; atrial fibrillation, atrial flutter, supraventricular premature beats (airv), left atrial enlargement, right atrial enlargement, right ventricular enlargement, ventricular extrasystoles, isolated ventricular extrasystoles, first-degree atrioventricular block, second-degree atrioventricular block, incomplete right bundle branch block, complete right bundle branch block, incomplete left bundle branch block, complete left bundle branch block, left anterior hemiblock, and left posterior hemiblock.

### Ethical aspects

This study received approval from the Institutional Review Board of the Evandro Chagas National Institute of Infectious Disease in September 2022 (Protocol No. 61674422.7.0000.5262). All investigators are committed to maintaining the confidentiality and privacy of participants' data. Study results will be disseminated through scientific channels, ensuring the anonymity of all participants. No conflicts of interest exist among the researchers or collaborators, and no restrictions have been placed on the public dissemination of findings, regardless of their concordance with the initial hypotheses. Inclusion criteria require that all participants provide written informed consent, adhering to *Good Clinical Practice (GCP) guidelines*. Patients will be instructed to read – or, when necessary, listen to the consent form without time constraints and all questions regarding the study's aims and procedures will be clearly addressed. Participants will be informed that enrollment is voluntary, and refusal to participate will not affect the quality or continuity of their treatment at the institution. Furthermore, they will be made aware of their prerogative to withdraw from study at any point without any penalty or consequence.

### Discussion

Although no clinical studies have specifically investigated the impact of AMIO treatment on mortality reduction in patients with CCC, some research suggests a beneficial effect of this drug on the survival rates of patients with ventricular arrhythmias of other etiologies. In the 1990s, the Argentine GESICA study (Randomized Trial of Low-dose Amiodarone in Severe Congestive Heart Failure), and the North American CHF-STAT study (Congestive Heart Failure: Survival Trial of Antiarrhythmic Therapy) evaluated whether the treatment with AMIO in patients with congestive heart failure affected survival

[32,33]. The GESICA trial reported that AMIO was effective in reducing mortality in individuals with congestive heart failure (CHF) [32]. In contrast, the CHF-STAT trial showed that although AMIO suppressed arrhythmias and improve ventricular function, it did not reduce the incidence of sudden death or improved overall survival [33].

The main reason suggested for these discrepant results was the differing proportions of patients with ischemic versus non-ischemic heart disease: 39% and 61% in GESICA compared to 71% and 29% in CHF-STAT, respectively. Furthermore, nearly 10% of the patients in the GESICA trial had CCC, whereas this condition was absent in CHF-STAT. These differences raise the possibility that AMIO may have a greater impact on reducing mortality in individuals with CCC [34].

Additionally, the BENEFIT study (2015) evaluated the effect of BZN in chronic patients with CCC and concluded that, although the treatment reduced the detection of *T. cruzi* in the blood, it did not reverse or prevent the progression of Chagas heart disease. Notably, the subgroup of patients receiving a combination of AMIO and BZN was the only one to show a reduction in hospitalizations and risk of cardiovascular death [35]. This clinical observation from a randomized clinical trial was further supported by animal model studies. In a mouse model of CD, the BZN/AMIO combination was more effective in reducing inflammation and cardiac fibrosis than either monotherapy [36,37]. These complementary preclinical and clinical findings strengthen the rationale and provide robust support for our hypothesis that AMIO may exert immunomodulatory and trypanocidal effects in humans with CCC, which is the focus of the present study.

The benefits of AMIO treatment in patients with CCC may be multifactorial. In addition to its antiarrhythmic effects, AMIO also exhibits other pharmacological properties well described in the literature, such as (i) antioxidant activity [38,39]; (ii) anti-inflammatory effects [40,41]; and (iii) α and β adrenergic antagonism [42]. Furthermore, AMIO has demonstrated both *in vitro* and *in vivo* activity against *T. cruzi* [16,36,37,43].

In patients with CCC, trypanocidal activity associated with AMIO treatment has not yet been observed. A retrospective study comparing patients treated with AMIO to controls did not demonstrate a significant antiparasitic effect [44]. Also, previous AMIO use was not associated with a positive PCR for *T. cruzi* under univariate or multivariate analyses [45]. However, these studies had limitations, such as the small number of patients treated with AMIO (only 37, and 38, respectively). In another clinical study, AMIO treatment in CCC patients led to a reduction in serum inflammatory cytokines (IFN-γ, TNF-α, IL-2, IL-4, and IL-6), suggesting a potential immunomodulatory effect of this drug, that may contribute to improved prognosis [26].

Although primarily used as an antiarrhythmic agent, AMIO may also exert potential immunomodulatory and trypanocidal effects that could provide therapeutic benefits in Chagas cardiomyopathy; however, its use is limited by well-defined adverse effects. Careful monitoring is required due to known toxicities, including thyroid dysfunction, hepatotoxicity, and pulmonary fibrosis, which should be considered alongside the risks of conventional therapies such as benznidazole. This exploratory study will evaluate whether the immunomodulatory and trypanocidal effects of AMIO correlate with improvements in echocardiographic and electrocardiographic dysfunctions caused by *T. cruzi* infection, potentially informing the development of new therapeutic strategies for patients with CCC.

## Acknowledgments

The authors thank the Dr. Solange Lisboa de Castro for critical reading of the manuscript.

## Author contributions

**Conceptualization:** Juliana Magalhães Chaves Barbosa, Alejandro Marcel Hasslocher-Moreno, Mauro Felippe Felix Mediano, Roberto Magalhães Saraiva, Anissa Daliry, Henrique Horta Veloso, Kelly Salomão.

**Data curation:** Juliana Magalhães Chaves Barbosa.

**Formal analysis:** Juliana Magalhães Chaves Barbosa, Amanda Faier-Pereira, Alejandro Marcel Hasslocher-Moreno, Evelyn Nunes Goulart da Silva Pereira, Mauro Felippe Felix Mediano, Roberto Magalhães Saraiva, Otacílio da Cruz Moreira.

**Methodology:** Juliana Magalhães Chaves Barbosa, Amanda Faier-Pereira, Evelyn Nunes Goulart da Silva Pereira, Otacílio da Cruz Moreira.

**Project administration:** Alejandro Marcel Hasslocher-Moreno, Mauro Felippe Felix Mediano, Roberto Magalhães Saraiva, Anissa Daliry, Henrique Horta Veloso, Kelly Salomão.

**Resources:** Alejandro Marcel Hasslocher-Moreno, Andréa Rodrigues Costa, Fernanda de Souza Nogueira Sardinha Mendes, Gilberto Marcelo Sperandio da Silva, Luiz Henrique Conde Sangenis, Marcelo Teixeira Holanda, Mauro Felippe Felix Mediano, Roberto Magalhães Saraiva, Sérgio Salles Xavier, Otacílio da Cruz Moreira, Anissa Daliry, Henrique Horta Veloso, Kelly Salomão.

**Supervision:** Anissa Daliry, Henrique Horta Veloso, Kelly Salomão.

**Writing – original draft:** Juliana Magalhães Chaves Barbosa.

**Writing – review & editing:** Juliana Magalhães Chaves Barbosa, Amanda Faier-Pereira, Alejandro Marcel Hasslocher-Moreno, Andréa Rodrigues Costa, Evelyn Nunes Goulart da Silva Pereira, Fernanda de Souza Nogueira Sardinha Mendes, Gilberto Marcelo Sperandio da Silva, Luiz Henrique Conde Sangenis, Marcelo Teixeira Holanda, Mauro Felippe Felix Mediano, Roberto Magalhães Saraiva, Sérgio Salles Xavier, Otacílio da Cruz Moreira, Anissa Daliry, Henrique Horta Veloso, Kelly Salomão.

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
