## [Editor Report · Decision Letter 0]

10 Sep 2025

Dear Dr. Salomão,

Thank you for submitting your manuscript to PLOS ONE. After careful consideration, we feel that it has merit but does not fully meet PLOS ONE’s publication criteria as it currently stands. Therefore, we invite you to submit a revised version of the manuscript that addresses the points raised during the review process.

We look forward to receiving your revised manuscript.

Kind regards,

Nicolas Padilla-Raygoza, Doctorate in Sciences

Academic Editor

PLOS ONE

Journal Requirements:

4. Please ensure that you refer to Figure 1 in your text as, if accepted, production will need this reference to link the reader to the figure.

**Additional Editor Comments:**

I consider that the protocol corresponds to experimental design (the effect of AMIO on CCC in Chagas Disease). This involves registering the protocol on a registry of clinical trials. Also, in the protocol is not include a sample of informed consent to verify if the consent includes the destiny of blood samples after the study.

---

## [Author Response · Author response to Decision Letter 1]

15 Sep 2025

PLOS ONE Decision:

Revision required [PONE-D-25-46528]

Dear Editor,

We thank you for the opportunity to revise and resubmit our manuscript entitled “Protocol of a Mixed (Retrospective and Prospective) Longitudinal Observational study of Amiodarone for Chronic Chagas Cardiomyopathy (AMIO-CHAGAS study): Investigating Immunomodulatory and Trypanocidal Effects ”. In response to the journal’s request:

Journal Requirements:

The manuscript has been revised in accordance with the PLOS ONE guidelines.

2. Please include your tables as part of your main manuscript and remove the individual files.

We have included the table as part of the main manuscript and removed it from the individual file.

3. Your ethics statement should only appear in the Methods section of your manuscript.

The ethics statement is included only in the Methods section.

4. Please ensure that you refer to Figure 1 in your text as, if accepted, production will need this reference to link the reader to the figure.

We apologize for the mistake; Figure 1 has been included in the manuscript text.

Response to the Editor Comments

I consider that the protocol corresponds to experimental design (the effect of AMIO on CCC in Chagas Disease). This involves registering the protocol on a registry of clinical trials. Also, in the protocol is not include a sample of informed consent to verify if the consent includes the destiny of blood samples after the study.

We appreciate the editor’s comments and the opportunity to clarify these points. We would like to emphasize that the submitted study is an observational clinical study, which includes both retrospective and prospective components. Unlike a clinical trial, our study does not involve the assignment of interventions, randomization of participants, or experimental manipulation of treatment. Instead, it aims to observe and analyze clinical outcomes under routine care conditions.

Because of these fundamental differences, the study does not meet the standard definition of a clinical trial, and therefore registration in a clinical trial registry is not applicable. Observational studies of this nature follow patients and collect data according to standard clinical practice rather than testing the efficacy or safety of an intervention under experimental conditions.

In summary, the study fully complies with ethical standards and methodological requirements for observational research, while not fitting the criteria for a clinical trial registry.

While revising your submission, please upload your figure files to the Preflight Analysis and Conversion Engine (PACE) digital diagnostic tool, https://pacev2.apexcovantage.com/.

As recommended, we uploaded the figures to PACE during the manuscript revision process.

---

## [Decision Letter · Decision Letter 1]

25 Feb 2026

Dear Dr. Salomão,

Thank you for submitting your manuscript to PLOS ONE. After careful consideration, we feel that it has merit but does not fully meet PLOS ONE’s publication criteria as it currently stands. Therefore, we invite you to submit a revised version of the manuscript that addresses the points raised during the review process.

We look forward to receiving your revised manuscript.

Kind regards,

Nicolas Padilla-Raygoza, Doctorate in Sciences

Academic Editor

PLOS One

Journal Requirements:

**Additional Editor Comments:**

The authors should answer all the comments from reviewers and resubmit a new version higlighted the changes in the manuscript.

Reviewers' comments:

Reviewer's Responses to Questions

**Comments to the Author**

1. Does the manuscript provide a valid rationale for the proposed study, with clearly identified and justified research questions?

Reviewer #1: Yes

Reviewer #2: Yes

2. Is the protocol technically sound and planned in a manner that will lead to a meaningful outcome and allow testing the stated hypotheses?

Reviewer #1: Yes

Reviewer #2: Partly

3. Is the methodology feasible and described in sufficient detail to allow the work to be replicable?

Reviewer #1: Yes

Reviewer #2: No

4. Have the authors described where all data underlying the findings will be made available when the study is complete?

The PLOS Data policy requires authors to make all data underlying the findings described in their manuscript fully available without restriction, with rare exception, at the time of publication. The data should be provided as part of the manuscript or its supporting information, or deposited to a public repository. For example, in addition to summary statistics, the data points behind means, medians and variance measures should be available. If there are restrictions on publicly sharing data—e.g. participant privacy or use of data from a third party—those must be specified.requires authors to make all data underlying the findings described in their manuscript fully available without restriction, with rare exception, at the time of publication. The data should be provided as part of the manuscript or its supporting information, or deposited to a public repository. For example, in addition to summary statistics, the data points behind means, medians and variance measures should be available. If there are restrictions on publicly sharing data—e.g. participant privacy or use of data from a third party—those must be specified.

Reviewer #1: Yes

Reviewer #2: No

5. Is the manuscript presented in an intelligible fashion and written in standard English?

Reviewer #1: Yes

Reviewer #2: Yes

You may also provide optional suggestions and comments to authors that they might find helpful in planning their study.

Reviewer #1: The AMIO-CHAGAS study protocol is robust, ethically sound, and of high clinical importance. It addresses an underexplored yet highly relevant topic in the management of chronic Chagas cardiomyopathy. The combination of immunological, molecular, and clinical endpoints represents an innovative and comprehensive approach.

Recommendations for Minor Improvement

Clarify exploratory of confirmatory analyses:

Specify which molecular and immunological measures (e.g., cytokines, gene expression, T. cruzi DNA) are confirmatory endpoints linked to the main hypothesis and which are exploratory analyses intended for hypothesis generation. This will improve methodological transparency and interpretability.

Describe long-term handling of biological samples:

Indicate how serum, plasma, and PBMCs will be managed after analysis—whether stored in a certified biobank, destroyed, or used for future ethically approved studies. Include brief details on storage duration and data governance.

Refine language and presentation:

A light English revision is suggested to enhance flow and precision:

Abstract: simplify and unify tenses.

Discussion: clarify links between preclinical findings and the study rationale.

Tables/Figures: define all abbreviations in legends.

Overall, the protocol is scientifically robust, ethically sound, and of high translational relevance. These minor refinements will improve clarity and publication readiness.

Reviewer #2: In its current form, the protocol addresses a relevant and potentially innovative research

question; however, it requires substantial methodological clarification, particularly regarding

the sample size calculation and the delimitation of the clinical scope of the proposed findings

.

Reviewer #1: **Yes:** MSc PhD Rubi Gamboa-LeónMSc PhD Rubi Gamboa-LeónMSc PhD Rubi Gamboa-LeónMSc PhD Rubi Gamboa-León

Reviewer #2: No

---

## [Author Response · Author response to Decision Letter 2]

30 Mar 2026

Response to Reviewers - PONE-D-25-46528R1

Protocol of a Mixed (Retrospective and Prospective) Longitudinal Observational study of Amiodarone for Chronic Chagas Cardiomyopathy (AMIO-CHAGAS study): Investigating Immunomodulatory and Trypanocidal Effects

Reviewer #1: The AMIO-CHAGAS study protocol is robust, ethically sound, and of high clinical importance. It addresses an underexplored yet highly relevant topic in the management of chronic Chagas cardiomyopathy. The combination of immunological, molecular, and clinical endpoints represents an innovative and comprehensive approach.

Recommendations for Minor Improvement

Clarify exploratory of confirmatory analyses:

1. Specify which molecular and immunological measures (e.g., cytokines, gene expression, T. cruzi DNA) are confirmatory endpoints linked to the main hypothesis and which are exploratory analyses intended for hypothesis generation. This will improve methodological transparency and interpretability.

Response: We appreciate this suggestion. In the present study, all molecular and immunological measures—including cytokine levels, gene expression, and T. cruzi DNA—are planned as confirmatory endpoints, as they are directly related to the main hypothesis evaluating the immunomodulatory and trypanocidal effects of amiodarone in patients with CCC. These diverse methodologies were intentionally adopted to broaden the range of parameters analyzed, thereby increasing the robustness and interpretability of the results. No analyses are exploratory, and all are designed to address this central research question.

2. Describe long-term handling of biological samples:

Response: Thank you for this important suggestion. The request has been addressed, and the relevant information has been added to the Methods section under the subsection “Blood Collection and Isolation of Plasma, Serum and Peripheral Blood Mononuclear Cells (PBMCs)”. This subsection now describes the procedures for long-term storage, management, and disposal of biological samples.

3. Indicate how serum, plasma, and PBMCs will be managed after analysis—whether stored in a certified biobank, destroyed, or used for future ethically approved studies. Include brief details on storage duration and data governance.

Response: Thank you for this comment. As previously indicated, the requested information has been included in the Methods section under the subsection “Blood Collection and Isolation of Plasma, Serum and Peripheral Blood Mononuclear Cells (PBMCs)”. Biological samples will be stored under institutional responsibility for up to five years after study completion and subsequently discarded according to institutional biosafety regulations. During this period, samples may be used for additional subprojects subject to appropriate ethical approval.

Refine language and presentation:

4. A light English revision is suggested to enhance flow and precision:

Response: Thank you for this suggestion. The manuscript has been carefully revised for English language, grammar, and clarity. All changes have been highlighted in the revised text.

5. Abstract: simplify and unify tenses.

Response: Thank you for this helpful suggestion. The abstract has been revised to simplify sentence structure.

6. Discussion: clarify links between preclinical findings and the study rationale.

Response: We thank the reviewer for this valuable suggestion. The Discussion has been revised to more clearly highlight how the clinical observations from the BENEFIT study are supported by experimental evidence from murine models, thereby providing robust support for our hypothesis regarding the potential immunomodulatory and trypanocidal effects of amiodarone in patients with CCC.

7. Tables/Figures: define all abbreviations in legends.

Response: Thank you for this suggestion. All abbreviations in tables and figure legends have been defined accordingly in the revised manuscript.

Overall, the protocol is scientifically robust, ethically sound, and of high translational relevance. These minor refinements will improve clarity and publication readiness.

Reviewer #2: In its current form, the protocol addresses a relevant and potentially innovative research question; however, it requires substantial methodological clarification, particularly regarding the sample size calculation and the delimitation of the clinical scope of the proposed findings

Scientific Review Report

The manuscript presents an ambispective longitudinal protocol (retrospective and prospective), entitled AMIO-CHAGAS, which evaluates the use of amiodarone in patients with chronic Chagas cardiomyopathy treated at a referral center in Brazil. Considering that Chagas disease remains endemic in this country and that Chagas cardiomyopathy represents a significant cause of disability and cardiovascular morbidity and mortality, the topic is clinically relevant.

The study population consists of patients with confirmed diagnosis of Chagas disease. The primary objective of the protocol is to evaluate the potential immunomodulatory and trypanocidal effects of amiodarone, exploring whether, in addition to its recognized antiarrhythmic effect, the drug could have an etiological impact on Trypanosoma cruzi infection.

1. Originality and Relevance

The protocol addresses a pharmacological aspect that has not yet been fully elucidated in humans: the potential immunomodulatory and trypanocidal effects of amiodarone beyond its conventional antiarrhythmic indication. Given that the available clinical evidence regarding this effect remains limited, the study aims to contribute to a relevant gap in current knowledge.

Furthermore, considering the persistent prevalence of Chagas disease and its impact in terms of cardiovascular morbidity and mortality, the exploration of additional effects of drugs already used in this population holds scientific and clinical interest.

2. Discussion and Clinical Applicability

Although the protocol proposes an innovative hypothesis regarding the potential etiological effect of amiodarone in humans, it would be advisable for the authors to more thoroughly contextualize the clinical relevance of these findings. In particular, it is important to discuss the drug's safety profile, characterized by known adverse effects, in comparison with currently established etiological therapies for Chagas disease. It is suggested that the clinical scope of the study be more clearly defined, emphasizing the exploratory nature of the research.

Response: We thank the reviewer for this valuable suggestion. The manuscript has been updated to address these points, and the relevant information regarding clinical relevance, safety profile, comparison with conventional therapies, and the exploratory nature of the study has been incorporated into the final paragraph of the Discussion.

3. Methodology

The methodological design is consistent with the stated objective; however, the proposed sample size (n = 90) may be limited for detecting clinically significant differences, particularly given the exploratory nature of the immunomodulatory and trypanocidal effects being evaluated. Although the authors report having based their estimate on two previous studies and increased the calculated sample size by 20% to account for potential losses, the parameters used for a formal sample size calculation are not described. The absence of this information makes it difficult to determine whether the study will have sufficient statistical power to support robust conclusions.

A more detailed description of the sample size calculation procedure would strengthen the methodological rigor of the protocol.

Response: We thank the reviewer for this important comment. We have revised the Methods section to provide additional details regarding the sample size calculation. Specifically, we included the significance level (α), statistical power (1 − β), standard deviations, and the minimum differences considered clinically relevant for the primary and secondary endpoints. The sample size of 90 participants (45 per group) was determined based on the scenario requiring the largest sample, with an additional 20% to account for potential losses, ensuring sufficient statistical power to detect meaningful differences in the inflammatory and echocardiographic parameters evaluated.

4. Ethical Aspects

The manuscript states that the study was approved by an institutional ethics committee, including the corresponding approval number. It also specifies that all participants provided informed consent prior to inclusion, and the authors declare the absence of conflicts of interest.

Overall, the protocol appears to comply with fundamental ethical standards for research involving human participants.

5. Reporting and Data Availability

The manuscript does not include an explicit statement regarding data availability. It is recommended that a specific section be incorporated clarifying whether the data will be publicly available, accessible upon reasonable request, or subject to ethical restrictions.

Response: We thank the reviewer for this important comment. We have updated the “Data Collection” section of the manuscript to clarify that the data generated in this study will be made available upon reasonable request, subject to ethical and privacy considerations. This revision ensures compliance with PLOS ONE’s data sharing policy and provides transparency regarding data accessibility.

6. Writing and Clarity

The manuscript is clearly written and allows adequate understanding of the objective and methodological structure of the protocol. The organization is coherent and facilitates comprehension of the scientific rationale.

7. Final Recommendation

In its current form, the protocol addresses a relevant and potentially innovative research question; however, it requires substantial methodological clarification, particularly regarding the sample size calculation and the delimitation of the clinical scope of the proposed findings.

Note: I declare that Google Translate was used exclusively for language translation assistance. No artificial intelligence tools were used in the scientific analysis or evaluation of the manuscript.

---

## [Editor Report · Decision Letter 2]

1 Apr 2026

Protocol of a Mixed (Retrospective and Prospective) Longitudinal Observational study of Amiodarone for Chronic Chagas Cardiomyopathy (AMIO-CHAGAS study): Investigating Immunomodulatory and Trypanocidal Effects

PONE-D-25-46528R2

Dear Dr. Salomão,

We’re pleased to inform you that your manuscript has been judged scientifically suitable for publication and will be formally accepted for publication once it meets all outstanding technical requirements.

Kind regards,

Nicolas Padilla-Raygoza, Doctorate in Sciences

Academic Editor

PLOS One

Additional Editor Comments (optional):

The corrections in the manuscript answer all comment from reviewers.
---

## [Editor Report · Acceptance letter]

PONE-D-25-46528R2

PLOS One

Dear Dr. Salomão,

I'm pleased to inform you that your manuscript has been deemed suitable for publication in PLOS One. Congratulations! Your manuscript is now being handed over to our production team.

Kind regards,

on behalf of

Dr. Nicolas Padilla-Raygoza

Academic Editor

PLOS One